# ADEM as an Initial Presentation of SLE: A Case Report

**DOI:** 10.3390/reports7030053

**Published:** 2024-07-05

**Authors:** Yousuf Sherwani, Ayham Alsaab, Mohan Sengodan

**Affiliations:** 1Medical City Arlington Hospital, Arlington, TX 76015, USA; ayham.alsaab@medicalcityhealth.com (A.A.); mohan.sengodan@hcahealthcare.com (M.S.); 2School of Medicine, West Virginia University, Morgantown, WV 26506, USA; 3School of Arts and Sciences, University of Pennsylvania, Philadelphia, PA 19104, USA

**Keywords:** autoimmune diseases, demyelinating, brain, anti-dsDNA antibodies, rituximab, IVIG

## Abstract

Acute disseminated encephalomyelitis (ADEM) is an inflammatory demyelinating disorder of the white matter. The pathophysiology is thought to be immune-mediated as in most cases the condition follows an infection or triggering incident. More recent literature has demonstrated that there may be a link between autoimmune conditions and ADEM. Here we present a case of ADEM in a middle-aged woman with systemic lupus erythematosus that recovered well after treatment with corticosteroids and rituximab.

## 1. Introduction

Acute disseminated encephalomyelitis (ADEM) is an inflammatory demyelinating disorder of the white matter. The pathophysiology is thought to be immune mediated as in most cases the condition follows an infection or triggering incident. Clinical manifestations include encephalopathy and rapid neurological decline over the course of days to weeks [1]. More recent literature has demonstrated that there may be a link between autoimmune conditions and ADEM [2]. ADEM is rare disease that is primarily present in children, with a prevalence of about one in 100,000 children being affected. Additionally, up to 85% of cases of ADEM have identifiable infection at the time of diagnosis or were recently vaccinated [3]. This case is unique as a middle-aged female was diagnosed with ADEM with no known triggering event such as infection or vaccination. Here we present a case of ADEM in a middle-aged woman with systemic lupus erythematosus that recovered well after treatment with corticosteroids and rituximab.

## 2. Detailed Case Description

A 37-year-old female presented to our hospital due to new onset severe headaches and right sided weakness with facial droop. The patient did not have any significant past medical history before this point. The patient was unable to speak properly and was found to have decreased responsiveness at the bedside. An initial CT scan of the head without contrast revealed an ill-defined edema involving the left periatrial white matter, the central gray matter and the external limited internal capsule without evidence of occlusion or hemorrhage. An MRI of the head with T2 Flair revealed diffuse edema and inflammation throughout the brainstem, pons, midbrain, and basal ganglia (Figure 1, Figure 2, Figure 3 and Figure 4). The T2 flair images revealed characteristic findings of ADEM such as tumefactive regions, asymmetric lesions, and lesions in the subcortical regions of the brain.electroencephalogram (EEG) showed diffuse slowing suggestive of moderate diffuse cerebral dysfunction without evidence of seizures or epileptiform activity. A lumbar puncture demonstrated a normal opening pressure, with cerebrospinal fluid containing 2 red blood cells/mm^3^, 9 white blood cells/mm^3^ of which 94% were lymphocytes, a glucose level of 58 mg/dL, and a protein level of 42 mg/dL. Both IgA and IgG immunoglobin were within normal limits in the CSF. Other proinflammatory cytokines in the CSF were not obtained at this admission. CSF bacterial cultures were negative, along with no evidence of an active herpes simplex virus or Epstein-Barr virus infection.Rapid plasma reagin was also negative. TSH was 1.3 mU/L and free T4 was 1.4 ng/dL; pulse dose steroids were initiated in the intensive care unit The patient was slow to improve and was transferred to the neurological intensive care unit for further management. After 17 days in the ICU, the patient was discharged to a rehabilitation facility. The patient slowly improved with physical therapy but did continue to have some refractory lower extremity weakness. At the time of discharge, our patient was discharged on 20 mg prednisone daily. An outpatient brain biopsy of the right frontal cortex revealed moderate gliosis but no abnormal lymphocytic infiltration. After thorough review of imaging findings and clinical presentation, a diagnosis of acute disseminated encephalomyelitis was made by Neurology.

This patient was referred to rheumatology due to the inflammatory findings seen on the head MRI with T2 FLAIR. Initial laboratory testing revealed a C-reactive protein level of 7.3 mg/dL (normal < 8 mg/dL), an ESR of 17 mm/h (normal 20 mm/h), ribonucleoproteins (RNP) negative, p-Antineutrophilic cyctoplasic antibody (ANCA) negative, c-ANCA negative, myeloperoxidase antibody negative, proteinase-3 antibody negative, SSA negative, SSB negative, RNP negative, C3 144 mg/dL (normal > 84 mg/dL), C4 50 mg/dL (normal > 14 mg/dL) antinuclear antibody (ANA) 1:80 (normal < 1:80), and an anti-dsDNA antibody level of 22 IU/mL (normal < 10 IU/mL) obtained through enzyme- linked immunosorbent assay (ELISA). Upon examination, the patient was found to have synovitis in the proximal interphalangeal joints of the hands bilaterally that was associated with morning stiffness. A diagnosis of systemic lupus erythematosus SLE was made according to the 2019 EULAR/ACR criteria [4]. The patient scored a 12 on the EULAR/ACR criteria due to her synovitis and positive anti-dsDNA. She did have clinical symptoms and thus met the criteria for the diagnosis of SLE. The patient was started on disease modifying antirheumatic drugs including methotrexate and hydroxychloroquine which she did not tolerate due to bleeding and anemia. Prednisone 20 mg daily was continued in an attempt to reduce flares of ADEM. Unfortunately, flares of ADEM continued which presented as lower extremity weakness, headaches, and facial droop in the months following diagnosis of ADEM. Subsequently, the patient was started on rituximab 10,000 mg every 4 weeks in an attempt to manage flares and improve symptoms. At the 6-month follow-up, the patient’s weakness and headaches were greatly improved. However, some residual left lower extremity weakness persisted which necessitated the use of a walker. Bloodwork revealed an anti-dsDNA antibody of 16 IU/mL at the 6-month follow-up visit. A follow-up MRI of the head with FLAIR revealed a marked decrease in the T2 Flair signal within the basal ganglia and midbrain with complete resolution of enhancement in the brainstem. During this time, neurology also recommended that the patient start Intavenous Immunoglobulins (IVIG) at 1 g/kg every 3 weeks. Gradual improvement in symptoms was seen over the next several months on a regiment of prednisone 15 mg daily, rituximab 1000 mg every four weeks, and IVIG 1 g/kg every 3 weeks. During her one year follow-up post-diagnosis, the patient had minimal residual weakness in her lower extremities and was able to ambulate without a walker. At this visit, her anti-dsDNA antibody continued to drop and was measured to be 12 IU/mL (Table 1). The patient has continued this regiment of prednisone, rituximab, and IVIG in an effort to avoid flairs of ADEM. She has endorsed a significant improvement in symptoms and quality of life. We appreciate the patient giving consent for us to publish her case.

## 3. Discussion

ADEM is thought to be an autoimmune disease in which it is theorized that autoimmune constituents attack myelin components such as myelin basic protein, proteolipid protein, and myelin oligodendrocyte protein leading to demyelination and diffuse white matter changes [5]. Demyelination is theorized to be precipitated by T cell activation through a cascade of inflammation involving inflammatory cytokines including tumor necrosis factor α (TNFα), interleukin 2 (IL2), and interferon γ (INFγ) [6]. Due to the active role of inflammatory cytokines in the pathogenesis of ADEM, any disease contributing to systemic formation of inflammatory cytokines can potentially be an etiologic factor for the initiation of ADEM.

In SLE, the number of T-cells, T helper type 1 cytokines, and other inflammatory cytokines such as TNFα increase substantially. The role of TNFα is especially important, since it exerts multiple stimulatory effects on T cells [7]. This inflammatory cascade leads to an altered permeability of the brain–blood barrier, which in turn allows the presentation of autoantigens to the activated immune system. This may lead to the autoimmune disease ADEM [2]. Migratory immune cells attack the basic myelin protein, and the final result is the demyelination seen in ADEM. 

A wealth of proinflammatory factors, including monocyte chemotactic protein 1, TNF-α, IL-1β, IL-6, and IL-8 are overexpressed with stimulation with anti-dsDNA antibodies. Accumulation of inflammatory cytokines is sufficient for accelerating the recruitment of immune cells and the induction of inflammatory processes. The anti-dsDNA antibodies can gain access to the brain tissue after the destruction of the blood–brain barrier, leading to inflammation and neuronal cell death [8]. Anti-dsDNA level is correlated with systemic lupus erythematosus activity [9]. We did not exclude other causes of autoimmune encephalitis such as anti-N-methyl-d-aspartate (NMDA) receptor encephalitis or glutamic acid decarboxylase (GAD) antibody-related encephalitis during the patient’s initial presentation. We have included a list of differential diagnosis in Table 2 that should be investigated when a patent presents with symptoms of ADEM [4].

Although our patient did not have any identifiable viral illness or symptoms to suggest viral illness, it is possible that she may have had asymptomatic viral infection that triggered her ADEM. Only about 50–85% of ADEM cases have an identifiable trigger such as viral illness or vaccination [4]. Regardless of whether our patient’s ADEM was triggered by an unidentified viral illness or was idiopathic in nature, we conclude that her underlying SLE exacerbated her pathology. In our patient, a reduction in anti-dsDNA antibodies was correlated with an improvement of symptoms and a reduction in the number of flares. There has been growing evidence of autoimmune disease associated with ADEM in patients with systemic lupus erythematosus, vasculitis, and rheumatoid arthritis [10]. We suggest the screening of all patients with ADEM for rheumatologic conditions. Immediate treatment should be initiated if autoimmune disease is diagnosed. Further clinical and experimental research will have to be performed to understand the underlying link between ADEM and systemic lupus erythematosus.

## Figures and Tables

**Figure 1 reports-07-00053-f001:**
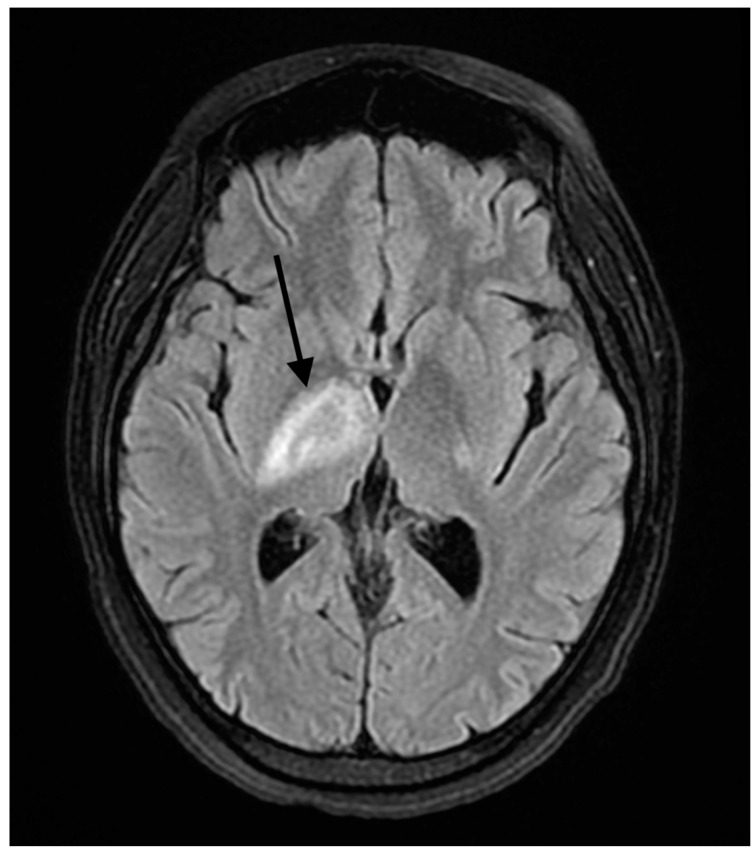
Fluid -attenuated inversion recovery (FLAIR) axial image of white matter lesions in the internal capsule. The arrow shows areas high signal changes consistent with edema and/or inflammation.

**Figure 2 reports-07-00053-f002:**
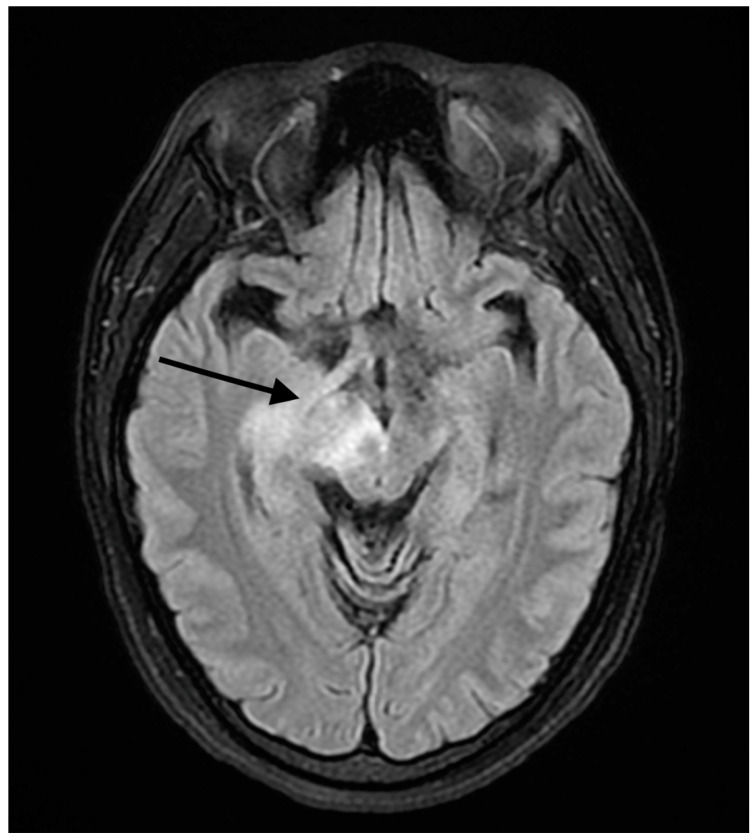
FLAIR axial MRI image reveals white matter lesions in the midbrain. The arrow shows areas high signal changes consistent with edema and/or inflammation.

**Figure 3 reports-07-00053-f003:**
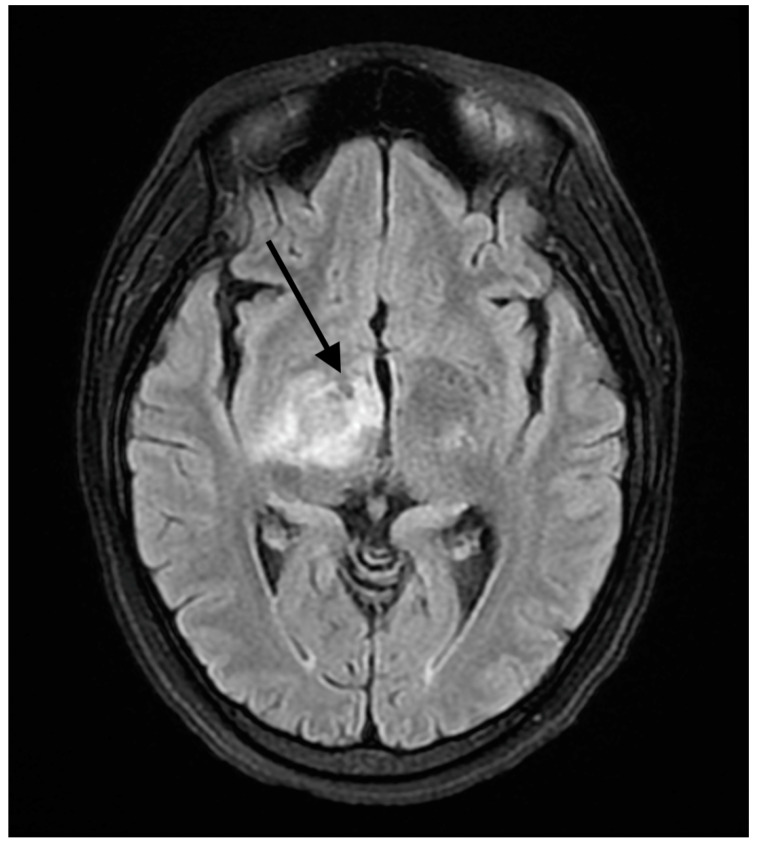
FLAIR axial MRI image reveals white matter lesions in the basal ganglia. The arrow shows areas high signal changes consistent with edema and/or inflammation.

**Figure 4 reports-07-00053-f004:**
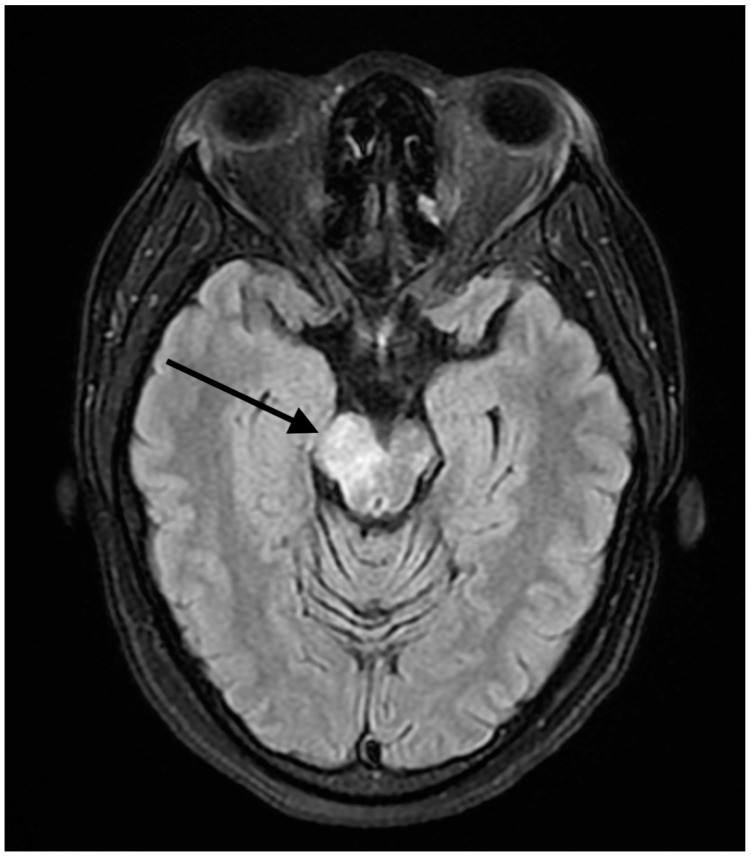
FLAIR axial MRI image reveals white matter lesions in the pons. The arrow shows areas high signal changes consistent with edema and/or inflammation.

**Table 1 reports-07-00053-t001:** Timeline of treatments offered and Symptoms present.

Days Since Establishing with Rheumatology	Treatment	Anti-dsDNA(IU/mL)	Symptoms
Day 1	Prednisone 20 mg	22	Bilateral lower extremity weakness and facial droop
Day 180	Prednisone 15 mg and Rituximab 1000 mg q4 week	16	Residual left lower extremity weakness. Resolution of facial droop
Day 360	Prednisone 15 mg, Rituximab 1000 mg q4 week, and IVIG q3 weeks	12	Resolution of lower extremity weakness and facial droop

**Table 2 reports-07-00053-t002:** Differential Diagnosis of ADEM.

Differential Diagnosis	Characteristics
Acute viral encephalitis	Fever, neck rigidity, and elevated acute phase reactants are typical.
Multiple sclerosis	First episode includes multifocal neurological deficits. Subsequent are usually not associated with acute encephalopathy.
Myelin oligodendrocyte glycoprotein disease (MOGAD)	More typical in children. Optic neuritis, spinal cord symptoms, and demyelination similar to ADEM may be present.
Metastatic brain cancer	Headache, seizures, and stroke-like symptoms present. Symptoms varied according to lesion location.
Neurobrucellosis	Varied symptoms. Headaches, fever, and neck stiffness most common. Hearing loss, confusion, and possible extremity weakness can also be present.
Cardioembolic stroke	Sudden numbness or weakness involving the face and/or extremities is typical. Confusion headaches and dysphagia are also common.
Cavernous sinus syndrome	Severe headaches, confusion, coma, seizures, vision loss, diplopia, and ptosis are common manifestations.
Cerebral venous thrombosis	Severe headaches, confusion, coma, seizures, vision loss, diplopia, and ptosis are common manifestations.
NDMA receptor encephalitis	Rapid progression of neurocognitive symptoms including changes in behavior, seizures, coma, dysarthria, and autonomic dysfunction are typical.

## Data Availability

The data presented in this study are available on request from the corresponding author due to privacy restrictions.

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
