# Peer review of "ADEM as an Initial Presentation of SLE: A Case Report"

_reports, 2024, doi:10.3390/reports7030053_

Round 1
Reviewer 1 Report
Comments and Suggestions for Authors
An Interesting Case Description
A case of autoimmune acute disseminated encephalomyelitis in an adult female with systemic lupus erythematosus.
This report provides a clear description of diagnostic methods, investigations for other diseases, and treatment approaches.
Author Response
Comment 1:
An Interesting Case Description
A case of autoimmune acute disseminated encephalomyelitis in an adult female with systemic lupus erythematosus.
This report provides a clear description of diagnostic methods, investigations for other diseases, and treatment approaches.
Answer 1: Thank you for your kind evaluation. We hope to publish this report soon.
Reviewer 2 Report
Comments and Suggestions for Authors
Dear Authors!
Thank you for the opportunity to review your manuscript.
In the introduction, you mentioned the virus etiology but in your case, no specific viruses were identified and ADEM was the first presentation of possible SLE. How did you discriminate ADEM from the component of SLE? This is a major concern.
The second major concern about the diagnosis of SLE: please clearly indicate which criteria you applied in 1997 or 2019, provide the reference, and give the number of diagnostic criteria. The diagnosis of SLE is very doubtful and borderline (slight positive ANA and slight positive anti-dsDNA).
Did you exclude other autoimmune encephalitides (NMDA? GAD? Hashimoto encephalitis?)? Please provide the data about the thyroid gland, status, and antithyroid antibodies.
All figures need arrows.
lines 64-65 "C3 144 mg/dL (normal < 193 mg/dL), C4 50 mg/dL (normal <57 mg/dL)". I am unsure that you provided the correct reference for C3 and C4. For SLE we are interested in the lower limit to detect hypocomplementemia. Here you provided the normal C3 and C4 levels. Give the lower limit of the reference
Please provide the reference for ANA 1:80
Please provide the table with differential diagnostics with short characteristics of each disease
Author Response
Comment 1 : In the introduction, you mentioned the virus etiology but in your case, no specific viruses were identified and ADEM was the first presentation of possible SLE. How did you discriminate ADEM from the component of SLE? This is a major concern.
Answer 1 : We discriminated ADEM from lupus encephalitis due primarily due to the MRI T2 Flair images and clinical presentation. The T2 flair images reveal characteristic findings of ADEMS such as tumefactive regions, asymmetric lesions, and lesions subcortical area (thalamus, basal ganglia, etc..) ( I have added this information on lines 35-37). Neurology and Radiology also agreed that the T2 Flair images were consistent with ADEM. Additionally the presentation of the patient was atypical of lupus encephalitis as it is more common for patients to present with delirium, psychosis, and seizures; all of which were absent in our patient. Additionally, although stroke may present in lupus encephalitis our patient had stroke like symptoms with no evidence of CVA on imaging. ( I have added some information on lines 33-34). Patient has more characteristic clinical presentation of ADEMS such as decreased responsiveness and severe headache (lines 28 and 30). Finally lumbar puncture did not reveal abnormal protein concentration or IgG concentration. (Line 40)
Comment 2: The second major concern about the diagnosis of SLE: please clearly indicate which criteria you applied in 1997 or 2019, provide the reference, and give the number of diagnostic criteria. The diagnosis of SLE is very doubtful and borderline (slight positive ANA and slight positive anti-dsDNA).
Answer 2: We used the 2019 EULAR/ACR classification criteria for systemic lupus erythematosus. We have included the reference in our submission. (Line 70) The diagnosis was made due to the patient having a positive anti-dsDNA with a weight of 6 and joint involvement including synovitis in 2 or more joints and atleast 30 minutes of morning stiffness with a weight of 6. The patient scored a 12 on the 2019 EULAR/ACR criteria. The patient was diagnosed with lupus due to scoring more than a 10 with atleast one clinical criteria being fulfilled. (I have included this discussion on lines 57-72).
Comment 3: Did you exclude other autoimmune encephalitides (NMDA? GAD? Hashimoto encephalitis?)? Please provide the data about the thyroid gland, status, and antithyroid antibodies.
Answer 3: Unfortunately we did not exclude NDMA or GAD encephalitis at the time of the patient’s initial presentation. I have included these limitations in the discussion section of the manuscript.(Lines 124-125) I have added TSH and T4 levels of the patient during her hospitalization. The thyroid function was found to be with in normal limits for this patient (line 42). Antithyroid antibodies were not considered necessary due to normal thyroid function.
Comment 4: All figures need arrows.
Answer 4: We have added arrows to aid viewers in finding white matter lesions in all figures.
Comment 5: lines 64-65 "C3 144 mg/dL (normal < 193 mg/dL), C4 50 mg/dL (normal <57 mg/dL)". I am unsure that you provided the correct reference for C3 and C4. For SLE we are interested in the lower limit to detect hypocomplementemia. Here you provided the normal C3 and C4 levels. Give the lower limit of the reference
Answer 5: We have adjusted the ranges provided for C3 and C4 so that the lower limit of normal is provided. Lines 67-68 are edited now to read “, C3 144 mg/dL (normal > 84 mg/dL), C4 50 mg/dL (normal > 14 mg/dL)”.
Comment 6: Please provide the reference for ANA 1:80
Answer 6: We have provided the reference range for ANA in line 68 so now it reads “ANA 1:80 (normal < 1:80)”
Comment 7: Please provide the table with differential diagnostics with short characteristics of each disease.
We have added Table 2 in the discussion section between lines 125 and 126 which includes differential diagnosis for ADEM along with Short characteristics.
Table 2.
Differential Diagnosis |
Characteristics |
Acute viral encephalitis |
Fever, neck rigidity, and elevated acute phase reactants are typical.
|
Multiple sclerosis |
First episode include multifocal neurological deficits. Subsequent usually not associated with acute encephalopathy.
|
Myelin oligodendrocyte glycoprotein disease (MOGAD) |
More typical in children. Optic neuritis , spinal cord symptoms, and demyelination similar to ADEM maybe present.
|
Metastatic brain cancer |
Headache, seizures, and stroke like symptoms present. Symptoms varied according to lesion location.
|
Neurobrucellosis |
Varied symptoms. Headaches, fever, neck stiffness most common. Hearing loss, confusion, and possible extremity weakness can also be present.
|
Cardioembolic stroke |
Sudden numbness or weakness involving face and/or extremities is typical. Confusion headaches and dysphagia also common.
|
Cavernous sinus syndrome |
Severe headaches, confusion, coma, seizures, vision loss, diplopia, and ptosis are common manifestations
|
Cerebral venous thrombosis |
Severe headaches, confusion, coma, seizures, vision loss, diplopia, ptosis are common manifestations
|
NDMA receptor encephalitis |
Rapid progression of neurocognitive symptoms including changes in behavior, seizures, coma, dysarthria and autonomic dysfunction are typical.
|
Reviewer 3 Report
Comments and Suggestions for Authors
This is an interesting clinical observation. I have some concerns which should be addressed adequately.
1. It is nice to change the title of the MS to "ADEM as an initial presentation of SLE: a case report".
2. How about inflammatory cytokines in the CSF in this patient? What is the reason for the onset of ADEM in this patient? This issue should be clarified. It is nice to conduct a literature review of similar cases.
3. The serum level of anti-dsDNA antibody in the patient is not so high. The measurement procedure for anti-dsDNA antibodies should be clarified (ELISA, RIA, etc).
4. Since the low titer of serum anti-dsDNA antibody, The rationale for the onset of ADEM in this patient should be discussed in more depth.
Author Response
Comment 1: 1. It is nice to change the title of the MS to "ADEM as an initial presentation of SLE: a case report".
Answer 1: We have changed the title in line 3 to ADEM as an initial presentation of SLE: a case report".
Comment 2: How about inflammatory cytokines in the CSF in this patient? What is the reason for the onset of ADEM in this patient? This issue should be clarified. It is nice to conduct a literature review of similar cases.
Answer 2: We have added information of inflammatory cytokines that were measured in the CSF. Line 41 was added which describes IgA and IgG immunoglobulins were within normal limits. We also have information about C3 and C4 levels on lines 67-68. The reason for the onset of ADEM in this patient remains unclear. Our patient did not have any identifiable viral illness or symptoms to suggest viral illness or recent vaccinations at time of presentation. It is possible that the patient had an asymptomatic viral illness before being admitted to our facility. Only about 50-85% of ADEM cases have an identifiable trigger such as viral illness or vaccination. Regardless of whether our patient’s ADEM was triggered by an unidentified viral illness or was idiopathic in nature, we conclude that her underlying SLE exacerbated her pathology. In our patient lower anti-dsDNA was correlated with improvement of symptoms. We have added this information on lines 127-133.
Comment 3: 3. The serum level of anti-dsDNA antibody in the patient is not so high. The measurement procedure for anti-dsDNA antibodies should be clarified (ELISA, RIA, etc).
Answer 3: The serum level of anti-dsDNA antibody was measured through ELISA. We have added this information on line 69 of the article.
Comment 4: Since the low titer of serum anti-dsDNA antibody, The rationale for the onset of ADEM in this patient should be discussed in more depth.
Answer 4: We have included the rationale for the onset of ADEM in this patient in lines 127-133. We have written “Although our patient did not have any identifiable viral illness or symptoms to suggest viral illness; it is possible that she may have had asymptomatic viral infection that triggered her ADEM. Only about 50-85% of ADEM cases have an identifiable trigger such as viral illness or vaccination.4 Regardless of whether our patient’s ADEM was triggered by an unidentified viral illness or was idiopathic in nature, we conclude that her underlying SLE exacerbated her pathology. In our patient lower anti-dsDNA was correlated with improvement of symptoms and reduction in the number of flares.”
Round 2
Reviewer 2 Report
Comments and Suggestions for Authors
Dear Authors!
Thank you for the revised version of the manuscript
I have no additional comments
Author Response
Comment 1: Dear Authors!
Thank you for the revised version of the manuscript
I have no additional comments
Answer 1: Thank you very much for the insightful review. We look forward to moving ahead with publication.
Reviewer 3 Report
Comments and Suggestions for Authors
The revised version is much improved.
If the authors did not exenined proinflammatory cytokines in the CSF, this issues should be added clearly in the text.
Author Response
Comment 1:
The revised version is much improved.
If the authors did not exenined proinflammatory cytokines in the CSF, this issues should be added clearly in the text.
Answer 1: We have added the information about not obtaining a panel of inflammatory cytokines in the CSF of the patient in lines 41-42. Thank you for your thoughtful review.